# Biological Basis and Computer Vision Applications of Image Phase Congruency: A Comprehensive Survey

**DOI:** 10.3390/biomimetics9070422

**Published:** 2024-07-10

**Authors:** Yibin Tian, Ming Wen, Dajiang Lu, Xiaopin Zhong, Zongze Wu

**Affiliations:** 1College of Mechatronics and Control Engineering, Shenzhen University, Shenzhen 518060, China; wenming@szu.edu.cn (M.W.); ludajiang@szu.edu.cn (D.L.); xzhong@szu.edu.cn (X.Z.); zzwu@szu.edu.cn (Z.W.); 2Guangdong Digital Economy and Artificial Intelligence Lab., Shenzhen 518060, China

**Keywords:** phase congruency, human visual system, Fourier analysis, wavelet transform, monogenic filter, image quality, feature detection, image registration, image fusion, object detection and tracking, deep neural network

## Abstract

The concept of Image Phase Congruency (IPC) is deeply rooted in the way the human visual system interprets and processes spatial frequency information. It plays an important role in visual perception, influencing our capacity to identify objects, recognize textures, and decipher spatial relationships in our environments. IPC is robust to changes in lighting, contrast, and other variables that might modify the amplitude of light waves yet leave their relative phase unchanged. This characteristic is vital for perceptual tasks as it ensures the consistent detection of features regardless of fluctuations in illumination or other environmental factors. It can also impact cognitive and emotional responses; cohesive phase information across elements fosters a perception of unity or harmony, while inconsistencies can engender a sense of discord or tension. In this survey, we begin by examining the evidence from biological vision studies suggesting that IPC is employed by the human perceptual system. We proceed to outline the typical mathematical representation and different computational approaches to IPC. We then summarize the extensive applications of IPC in computer vision, including denoise, image quality assessment, feature detection and description, image segmentation, image registration, image fusion, and object detection, among other uses, and illustrate its advantages with a number of examples. Finally, we discuss the current challenges associated with the practical applications of IPC and potential avenues for enhancement.

## 1. Introduction

Phase congruency is a fundamental concept in signal processing, particularly in image processing and computer vision. It involves the alignment of local phase components in a signal, which is crucial for the detection of image features such as edges, corners, and textures. Image Phase Congruency (IPC) was first introduced by Morrone and Owens in the mid-1980s as a means to mimic the human visual system’s ability to detect features based on phase alignment [1,2]. Since then, IPC has become a widely used tool for various image processing and computer vision tasks.

At its essence, IPC relies on the fact that significant image features coincide with locations where the Fourier components of an image are in phase with each other. This alignment suggests the presence of important features like edges or corners [3]. One of the key advantages of IPC is its ability to analyze images across multiple scales and orientations, enabling the detection of features of varying sizes and shapes. In computer vision, IPC has been proven to be invaluable for feature detection. For example, there have been a large number of traditional edge detection methods that rely on intensity differentiation, such as the Sobel, Canny, or second Derivatives of Gaussian (DoG) operators [4,5,6]. However, these methods can be sensitive to contrast variations and may struggle in complex scenes. Phase congruency offers an alternative that is more robust to these challenges [3,7,8].

Since the publication of Kovesi’s computational method for IPC in the 1990s [3], it has been widely used in a variety of applications in computer vision, ranging from low-level to high-level tasks. These applications include image denoise, image quality assessment, autofocus, image super-resolution, feature detection and description, image segmentation, image registration, image fusion, object detection and recognition, and so on [3,8,9,10,11,12,13,14,15,16,17]. Despite its effectiveness, one major drawback of IPC is its high computational cost, mainly due to its reliance on multi-scale analysis. However, advancements in semiconductor and computer chip technology in recent years have significantly reduced the impact of such computational burden. There has been intensive ongoing research and development to improve and enhance the utilization of IPC [18,19,20,21,22,23,24]. However, as far as we know, there has been no comprehensive review of its advancements since its inception almost four decades ago.

In this comprehensive survey, we explore the origins of IPC in human visual perception, compare its various computational implementations, and examine its diverse applications in both image processing and computer vision. Additionally, we discuss the challenges and potential enhancements needed for IPC to be more widely employed. For sake of clarity and conciseness, we do not distinguish between image processing and computer vision below.

## 2. Phase Congruency in Biological Perception

### 2.1. Frequency Analysis in Biological Perception

Human brains interpret and understand sounds, images, and other sensory inputs based on their constituent frequencies. For example, the human ear is sensitive to different frequencies of sound waves, and the brain interprets these frequencies to discern pitch, timbre, and other auditory qualities [25,26,27]. Music perception, in particular, relies heavily on frequency analysis, as melodies, harmonies, and rhythms are all composed of specific frequencies that the brain learns to recognize and appreciate. In the human visual system, it has been convincingly shown that frequency analysis relates to the perception of patterns and textures [28,29,30,31]. For instance, certain visual patterns might be composed of repeating elements that create a particular frequency or rhythm. The visual system is adept at recognizing these patterns and using them to interpret our surroundings. More broadly, frequency analysis in human perception involves the brain’s ability to extract meaningful information from sensory inputs based on their frequency content [27,32]. This process is fundamental to our understanding of and interactions with the world around us.

In the frequency domain, a signal is represented by a spectrum of frequencies, each with its own amplitude and phase. This spectral representation provides valuable insights into the signal’s characteristics. For instance, high-frequency components correspond to fine details, while low-frequency ones represent smoother parts.

### 2.2. Fourier Transform and Phase Congruency

Fourier transform is a well-established mathematical operation that decomposes a signal into its constituent frequencies. For a given signal, there are theoretically infinite constituent frequencies; thus, in practice, the Fourier transform is essentially an approximation that utilizes the weighted sum of a number of basis functions, each of which represents a constituent frequency [33]. Figure 1 illustrates a 1D pulse function and its approximations with the sum of different numbers of Fourier basis functions. It is obvious that the more basis function terms used, the more accurate the Fourier transform, as is the case in any approximation method. It is less intuitive but graphically conspicuous that at the two jumping steps of the pulse function shown as locations $P_{1}$ and $P_{2}$ in Figure 1, the Fourier components and their summations are perfectly aligned, as highlighted with the dashed circles. These two locations are where the phases of the Fourier components are in complete agreement and the signal reaches high phase congruency. This will be mathematically described below.

The Fourier transform of a 1D signal g(t) is as follows:(1)$$G_{n}(x)=\int_{-\infty }^{+\infty }g(t)e^{-i2\pi xt}dt=Re[G_{n}(x)]+Im[G_{n}(x)],$$
where n is the Fourier component number, while $Re[G_{n}(x)]$ and $Im[G_{n}(x)]$ are the real and imaginary components of $G_{n}(x)$. Its amplitude and phase are as follows:(2)$$A_{n}(x)=\left|G_{n}(x)\right|=\sqrt{{\left\{Re[G_{n}(x)]\right\}}^{2}+{\left\{Im[G_{n}(x)]\right\}}^{2}},\;\emptyset_{n}(x)=atan\left(\frac{Re[G_{n}(x))]}{\;Im[G_{n}(x))]}\right),$$

Phase congruency is defined as follows [2]:(3)$$PC(x)={MAX}_{\bar{\emptyset }(x)\in [0,2\pi ]}\left\{\frac{\sum_{n=1}^{N}A_{n}(x)cos[\emptyset_{n}(x)-\bar{\emptyset }(x)]}{\;\sum_{n=1}^{N}A_{n}(x)}\right\},$$
where $A_{n}$ represents the amplitude of the Fourier component n, $\emptyset_{n}(x)$ its local phase at position x, and N the total number of components. The value of $\bar{\emptyset }(x)$ is the amplitude weighted mean local phase of all the Fourier terms at the point under consideration. The term $cos[\emptyset_{n}(x)-\bar{\emptyset }(x)]$ is approximately equal to one minus half of ${\left[\emptyset_{n}(x)-\bar{\emptyset }(x)\right]}^{2}$ when it is small according to the Taylor expansion. The operation ${MAX}_{\bar{\emptyset }(x)\in [0,2\pi ]}\left\{f\right\}$ to find the maximum of f is equivalent to finding where the weighted variance of local phase is minimal relative to the weighted average local phase.

### 2.3. Biological Basis of Phase Congruency

As briefly mentioned in Section 2.1, both the human auditory and visual systems utilize frequency analysis for information processing. There are reasons to believe that different sensory systems may share some common operating mechanisms. For example, both vision and hearing employ hierarchical processing models [34,35]. Hermman et al. reported that synchronization of neural activity is more sensitive to regularity imposed by a coherent frequency modulation in sounds compared with the sustained response, which implies the role of phase congruency in some auditory functions [36]. In addition, studies on the patterns of cortical connectivity have found evidence for direct connections between different sensory primary cortices, especially hearing and vision, which suggests that the basis for cross-modal interactions to affect perceptual processing is present at very early stages of sensory processing [37,38]. Nevertheless, in this report we mostly focus on vision-related phase congruency. For structural simplicity, we divide this section into two parts, i.e., spatial phase congruency and temporal phase congruency. Though they are closely related, we mainly discuss the spatial aspect in this report.

#### 2.3.1. Spatial Phase Congruency

Phase congruency was initially proposed in the psychophysical study of visual phenomena. Mach bands are a visual illusion named after the 19th-century Austrian physicist Ernst Mach, who first described the phenomenon. It refers to the illusionary perception of brighter and darker bands along the boundaries where two different intensities meet [39]. That is, when a bright area is adjacent to a dark area, an even brighter band appears to exist along the edge of the bright area, and a darker band along the edge of the dark area, despite the fact that these bands are not physically present in the pattern. The illusion is not caused by differences in the physical intensity of light at the boundaries, but rather by the way the visual system processes these changes. One explanation is that Mach bands arise due to lateral inhibition, a neural process where excited neurons inhibit the activity of neighboring neurons [40]. However, this lateral inhibition theory cannot explain that there are no Mach bands in a square waveform [41]. Inspired by such observations, Morrone et al. demonstrated that phase relationships between Fourier components can explain the physically non-existing structure in their groundbreaking study on an odd and even symmetry visual field model [1]. As shown in Figure 2, there are no contrast variations in the intensity gradients of the step pattern across the stripes, but there are variations in the weighted local phase.

Burr et al. experimentally showed that the human visual system has phase response-based line and edge detectors, and the receptive fields are symmetric; one class is even-symmetric related to line detectors, the other odd-symmetric [42]. Based on these psychophysical observations, Morrone et al. developed a local energy model such that the local maxima of the local energy functions occur at locations of maximal phase congruency, and two-step simultaneous line and edge detectors based on the model, one step is linear and the other nonlinear [2,43,44,45]. Other researchers made various improvements to this class of local energy model-based feature detectors and also extended to 3D images [46,47].

More recently, physiological studies on primate and human brains provided additional evidence of phase congruency being employed by the visual system. Ringach conducted electrophysiological recording of simple cells in primate primary visual cortex (V1) and found Gabor-like receptive fields with odd-symmetry and even-symmetry [48]. Perna et al. showed sensitivity to phase congruency associated with edges and lines in human V1, but more importantly, they found that only higher-level areas can recognize phase types [49]. Henriksson et al. further employed functional magnetic resonance imaging (fMRI) to look for phase-sensitive neural responses in the human visual cortex and found sensitivity to the phase difference between spatial frequency components in all studied visual areas, all of which showed stronger responses for the stimuli with congruent phase structure [50]. Some results from this study are shown in Figure 3.

Computational studies also indicate that the detection of phase congruency probably involves higher-level visual cortex, even more than V1. For example, a natural image statistics study showed that pooling across multiple frequencies is statistically optimal to utilize the V1 output in V2 [51]. Thompson showed how image higher-order statistics can be modified so that they are sensitive to image phase structure only and that natural images have consistent higher-order statistical properties, differentiating them from random-phase images with the same power spectrum. Thus, it is possible that the sensitivity to relative phase can be determined directly by the higher-order structure of natural scenes [52].

#### 2.3.2. Temporal Phase Congruency

For a significant portion of the time the visual system is experiencing non-static stimuli. In other words, it receives a sequence of images instead of a static one, even if we ignore the microsaccades of the eye [53]. Some experimental evidence suggests that feature tracking may be used by the human visual system [54,55]. Frequency analysis has been widely applied to motion perception. The usage of spatiotemporal filter-based models has been well established; that is, motion detectors can be constructed as spatiotemporal filters [56,57,58]. The transformation of such filters from the frequency space to the space-time space makes them more intuitive, which results in the spatiotemporal receptive field, oriented in space-time [59].

It has been experimentally shown that photoreceptors exploit nonlinear dynamics to selectively enhance and encode local phase congruency of temporal stimuli, and that to mitigate for the inherent sensitivity to noise of the local phase congruency, the photoreceptor nonlinear coding mechanisms are tuned to suppress random phase signals [60]. Another study showed that temporal processing by photoreceptors alone, in the absence of any spatial interactions, improved target detection from cluttered background dramatically, which is also explained by photoreceptor temporal non-linear dynamic models [61]. These findings are consistent with the well-established theory that non-linear processing by the visual system is matched to the statistics of natural scenes [62,63].

Del Viva et al. reported a strong dependency of perception of motion transparency on the relative phase of harmonic components of one-dimensional gratings. A feature-tracking model computing local energy function from a pair of space-time separable front stage filters combined with a battery of directional second stage mechanisms is able to quantitatively emulate the phase congruency dependence illusion and the insensitivity to overall phase [64]. Fleet et al. proposed a method to compute the 2D component velocity from image sequences using the first-order behavior of surfaces of constant local phase, resulting in high-resolution and robust velocity detection in the presence of contrast, scale, orientation, and speed [65].

### 2.4. Image Phase Congruency and Phase Correlation

A closely related concept arising from Fourier analysis is phase correlation, which involves computing the Fourier transforms of two images and then calculating their cross-power spectrum.
(4)$$R_{PC}(x,y)=F^{-1}\left\{\frac{G_{a}\;(u,v)\;⨀\;{G_{b}}^{*}(u,v)}{\;\left|G_{a}\;(u,v)\;⨀\;{G_{b}}^{*}(u,v)\right|}\right\},$$
where $F^{-1}\left\{\cdot \right\}$ is the inverse Fourier transform; $G_{a}\;(u,v)$ and $G_{b}\;(u,v)$ are the Fourier transforms for input signals $g_{a}\;(u,v)$ and $g_{b}\;(u,v)$; ⨀  is the Hadamard product; ${G_{b}}^{*}(u,v)$ is the conjugate of $G_{b}\;(u,v)$; and (x,y) and (u,v) are the spatial coordinates and frequencies, respectively. The phase correlation contains information about the relative displacement between the two signals. By performing an inverse Fourier transform on the cross-power spectrum, a correlation function is obtained with a peak located at the relative displacement. The position of this peak indicates the relative shift between the two inputs. Phase correlation may be involved in certain perceptual processes, such as temporal correlations and feature integration [66], but studies are very limited, and the topic is not within the scope of this report.

Mathematically, both phase congruency and phase correlation use Fourier analysis to utilize the phase instead of the amplitude of images. Both involve computing Fourier transforms and manipulating phase values to extract meaningful information from signals and share the advantages of being insensitive to illumination and contrast variations, scales, orientations, and noises [3,67]. Phase-based methods’ insensitivities to image contrast play a critical role in many applications, as discussed in later sections. For example, phase congruency and phase correlation both reduce the impact of nonlinear radiometric differences due to nonlinearity in the imaging system, sensor characteristics, atmospheric conditions, or post-processing adjustments, which are frequently encountered in remote sensing and multi-modal image fusion [68].

One important distinction, as their definitions in Equations (3) and (4) illustrate, is that phase congruency reflects the local phase relationship of features within a single signal or image, while phase correlation indicates the spatial relationship between two similar signals or images. Another very important difference is that phase congruency is local, while phase correlation is global. Due to these differences, their applications in computer vision are also very different. Though both have been employed for image registration and motion detection [13,67,69], phase congruency essentially acts as local feature detectors and other feature-based processing is necessary, while phase correlation can directly provide the relative displacement between the two images as a global operator.

## 3. Computational Implementations of IPC

IPC is difficult to implement as originally defined by Morrone et al. (Equation (3)) [1]. For practical applications, various alternatives have been proposed for IPC since then [3,70,71,72]. The best known and most widely utilized implementation was by Kovesi [3], though it is not necessarily the most efficient one [71].

### 3.1. Relevant Computational Aspects for IPC Implementation

Fundamentally, IPC is a multiscale technique. Multiscale image analysis typically decomposes an image into multiple components, each corresponding to a different scale [73,74]. The decomposition can be carried out using Fourier transform, wavelet transform, principle or independent component analysis, curvelet transform, etc. [75,76,77,78]. The scales can range from very fine details to broader, more abstract patterns. At each scale, specific techniques are employed to extract relevant features or metrics. The extracted features from multiple scales are then combined or analyzed jointly to gain a comprehensive understanding of the image or dataset. The multiscale approach allows for the capture of both local and global information, enabling a more robust and accurate interpretation of the image. Understanding how IPC handles noise in images is another important computational aspect. Techniques to enhance noise robustness, such as preprocessing steps to reduce noise or post-processing steps to filter out spurious detection, are employed for IPC computation [79].

### 3.2. IPC Computation from FOURIER and Hilbert Transforms

Vanketash et al. proposed the first computationally friendly approach for IPC based on the finding that the points of maximum phase congruency are equivalent to peaks in the local energy function [70]. Wang et al. presented an implementation of IPC computation using 2D Hilbert transforms [80].
(5)$$E(x,y)=\sqrt{{\tilde{g}(x,y)}^{2}+{H[\tilde{g}(x,y)]}^{2}\;},$$
where $\tilde{g}(x)$ is the input image without the DC component, while H[⋅] is the Hilbert transform. The phase congruency is directly proportional to the local energy function.
(6)$$E(x,y)=PC(x,y)\sum_{n=1}^{N}A_{n}(x,y),$$
where PC(x,y) is the phase congruency, while N is the total number of Fourier components. Thus, phase congruency can be calculated by combining Equations (5) and (6).
(7)$$PC(x,y)=\frac{E(x,y)}{\sum_{n=1}^{N}A_{n}(x,y)}=\frac{\sqrt{{\tilde{g}(x,y)}^{2}+{H[\tilde{g}(x,y)]}^{2}\;}}{\sum_{n=1}^{N}A_{n}(x,y)},$$

It becomes clear that phase congruency is independent of the magnitude of the input signal. It is a unitless value within the range [0,1]. This relationship is graphically illustrated in Figure 4. This is a very important property, as it is insensitive to image illumination and contrast variations. Thus phase congruency has built-in amplitude normalization, which is a strategy widely used in image processing such as normalized cross-correlation and other more sophisticated methods [81,82].

As the Hilbert transform is a multiplier operation, in the frequency domain it has the effect of shifting the phase of the negative and positive frequency components by π/2 and −π/2, respectively [83]. Thus, for 2D images, Equation (7) can be easily implemented by discrete Fourier transform.

### 3.3. IPC Computation from Wavelet Transform

In the mid-1990s, Kosevi proposed to use wavelet transform to compute phase congruency. It was implemented as part of his PhD dissertation [84]. This seminal work and later improvement reached a much wider audience when it was published as journal papers and open-source computer code [3,7]. The majority of the work using IPC for computer vision applications has adopted this approach. To preserve the phase information, linear-phase filters are necessary. As such, non-orthogonal wavelets in even-symmetric and odd-symmetric quadrature pairs are utilized.
(8)$$\left[e_{n}(x),\;o_{n}(x)\right]=i(x)*\left[W_{n}^{e},W_{n}^{e}\right],$$
where i(x) is the input signal, $\left[W_{n}^{e},W_{n}^{e}\right]$ is the quadrature pair of even-symmetric and odd-symmetric filters at a scale n, and $\left[e_{n}(x),\;o_{n}(x)\right]$ is the output of the filter pair. More specifically, log-Gabor filters are used (illustrated in Figure 5), and the bandwidths of filters are set to constant at all scales. The corresponding amplitude and phase are as follows:(9)$$A_{n}(x)=\sqrt{{e_{n}(x)}^{2}+{o_{n}(x)}^{2}\;},\;\emptyset_{n}(x)=atan\left(\frac{e_{n}(x)}{\;o_{n}(x)}\right)$$

The local energy is as follows:(10)$$E(x)=\sqrt{{\left[\sum_{n=1}^{N}e_{n}(x)\right]}^{2}+{\left[\sum_{n=1}^{N}o_{n}(x)\right]}^{2}\;},$$

And similar to Equation (7), the phase congruency is expressed as follows:(11)$$PC(x)=\frac{E(x)}{\sum_{n=1}^{N}A_{n}(x)},$$

Computationally, Equation (11) is not stable if the filter outputs are all near zeros, which can be solved by adding a non-zero negligible value. In addition, to take into account noise in real signals and images, a noise compensation is added to the local energy term, which is equivalent to soft thresholding for denoise [85]. It should be noted that the distribution of filter responses should not be too narrow and that a uniform distribution is of particular significance as step discontinuities are common in images. A weighting function is added to devalue phase congruency at locations where the filter response spread is narrow. In addition, at a point of phase congruency, the cosine of the phase deviation should be large and the absolute value of the sine of the phase deviation should be small. Using their difference will increase phase congruency sensitivity. Taking all these factors into consideration, a practical computational definition of phase congruency is augmented as follows:(12)$$PC(x)=\frac{\sum_{n=1}^{N}\left\{W(x)\left\lfloor A_{n}(x)\Delta \emptyset_{n}(x)-T\right\rfloor \right\}}{\sum_{n=1}^{N}A_{n}(x)+\varepsilon },$$
where ε is the smallest non-zero value for the computing platform, T is the noise threshold, W(x) is the weight function, $\left\lfloor f\right\rfloor =f$ is for positive f, $\left\lfloor f\right\rfloor =$ 0 is for non-positive f, and Δ∅(x) is the phase deviation function, which is calculated as follows:(13)$$\Delta \emptyset_{n}(x)={cos[\emptyset }_{n}(x)-\bar{\emptyset }(x)]-\left|{sin[\emptyset }_{n}(x)-\bar{\emptyset }(x)]\right|,$$

For 2D images, Equation (12) can be expanded to the following expression:(14)$$IPC(x,y)=\frac{\sum_{m=1}^{M}\sum_{n=1}^{N}\left\{W_{m}(x)\left\lfloor A_{m,n}(x)\Delta \emptyset_{m,n}(x)-T_{m}\right\rfloor \right\}}{\sum_{n=1}^{N}A_{m,n}(x)+\varepsilon },$$
where M and N are the total numbers of orientations and scales, respectively. The spread weighting function in 2D $W_{m}(x)$ should be separable, and ideally it should be a 2D Gaussian function to minimize its impact on phase.

Obviously, a larger number of orientations and scales can produce more accurate IPC at the cost of more computations. However, today’s computing platforms support highly parallel processing, and if properly handled, the impact on processing speed can be significantly reduced.

### 3.4. IPC Computation from Monogenic Filters

Felsberg et al. proposed to generalize an analytical signal to two dimensions as a monogenic signal by using the Reisz transform and to use the monogenic filters termed Spherical Quadrature Filters (SQFs) to calculate the phase and orientation of such the signal [72,86]. This approach is more efficient than computing IPC from the wavelet transform described above.

The monogenic signal of an input g(x,y) is calculated as follows:(15)$$g_{M}(x,y)=[g(x,y),\;(h_{1}*g)(x,y),\;(h_{1}*g)(x,y)],$$
where $h_{1}(x,y)$ and $h_{2}(x,y)$ are the convolutional kernels of the Reisz transform. In spherical coordinates, the amplitude, local phase, and local orientation angles can be computed by the following expression:(16)$$A_{g}(x,y)=[\sqrt{{g(x,y)}^{2}+{\left(h_{1}*g)(x,y\right)}^{2}+{\left(h_{2}*g)(x,y\right)}^{2}\;}\;,$$
(17)$$g(x,y)=A_{g}(x,y)cos(\varphi ),\;\;\left(h_{1}*g\right)\left(x,y\right)=A_{g}(x,y)sin(\varphi )cos(\theta ),\;\left(h_{2}*g\right)\left(x,y\right)=A_{g}(x,y)sin(\varphi )sin(\theta ),$$
where φ∈[0,2π] and θ∈[0,π] are the local phase and local orientation, respectively. The triplet [$g,h_{1},h_{2}]$ forms a set of monogenic SQFs.

The outputs from the SQFs at two different scales can be used to calculate the cosine of the angle between these vectors by the scalar product and the absolute value of the sine by the magnitude of the cross product. And IPC can be computed as follows:(18)$$IPC(x,y)=\frac{cos(\varphi_{1}-\varphi_{1})}{|sin(\varphi_{1}-\varphi_{1})|+1}=\frac{f_{1}(x,y)\;\cdot \;f_{2}(x,y)}{\left|f_{1}(x,y)\times f_{2}(x,y)||f_{1}(x,y)|\;|f_{2}(x,y)\right|},$$

Wang et al. proposed an alternative implementation for IPC using monogenic filters and experimentally showed that it consumed lower time and smaller memory space than IPC from log-Gabor filters and that SQFs can not only overcome the limitations of log-Gabor filters but also improve the location accuracy and noise robustness with comparable or better performance [87]. It should be noted that Kovesi also adopted monogenic filters in his later computer code implementation [88].

### 3.5. A unified Formulation for IPC Computations

Forero et al. proposed a unified formulation for IPC from both log-Gabor filters and monogenic filters by expressing it as the product of three different components [71].
(19)IPC(x,y)=W(x,y)∗PCQ(x,y)∗NC(x,y),
where W(x,y) is the weight function based on frequency distribution, PCQ(x,y) the phase congruency quantification function, and NC(x.y) the noise compensation. The phase congruency quantification is the raw phase congruency computed as follows:(20)$$PCQ(x,y)=\left\lfloor 1-\alpha \left|\Delta \phi (x,y)\right|\right\rfloor ,$$
where α is a hyper parameter for sensitivity tuning, and $\Delta \phi \left(x,y\right)$, as illustrated in Figure 4b, is approximated by the phase deviation instead of its cosine in Equation (13). This makes IPC more sensitive to phase deviations.

For IPC from log-Gabor filters, the noise compensation term is as follows:(21)$$NC(x,y)=\frac{\left\lfloor E(x,y)-T\right\rfloor }{E(x,y)+\varepsilon },$$

And for IPC from monogenic filters, the following expression is used:(22)NC(x,y)=u[PCQ(x,y)],
where u[·] is the Heaviside step function [89].

### 3.6. IPC for 3D Images

Feature detection via the local energy model underlying IPC has been applied to 3D images by using 3D Morlet wavelets [47]. IPC has also been extended to 3D images by using conic spread filters as the weighting function [90]. It has been applied to confocal microscopy, seismic data analysis, and crack detection in materials [90,91,92,93], which will be discussed in more detail in the next section.

## 4. Applications of Phase Congruency in Low-Level Computer Vision

As illustrated in Figure 6, IPC has been utilized in different ways for various computer vision applications, which can be simplified as three different approaches or some of their combinations, i.e., (1) some of the intermediate results of computing IPC, including the scale-orientation feature maps and/or noise statistics thereof; (2) IPC itself; (3) image features, statistics, or transformations derived from IPC. For computer vision applications, we label them as low-, mid-, and high-level. It should be noted this classification is not rigorous, and they approximately correspond to the operations of image-to-image, image-to-features, and features-to-analysis, respectively. Due to the complexity of computer vision, real applications usually consist of multiple tasks, and some of them may involve different levels of operations. And as expected, IPC applications usually involve employment of other image processing and computer vision techniques. We will discuss each application category in detail in the following sections.

IPC has been widely adopted in various low-level computer visions, such as image denoise, image quality assessment, autofocus and blur detection, image super-resolution, and so on.

### 4.1. Image Denoise Using IPC

As noise is ubiquitous in real images, denoise is an essential low-level task in computer vision, usually as the first step of processing. The raw phase congruency, i.e., the phase congruency quantification function (the PCQ(x,y) in Equation (19)), is sensitive to noise. As such, for practical implementation, noise compensation (the NC(x,y) in Equation (19)) is utilized as the log-Gabor filters are suitable for noise detection [3]. Moreover, denoise is an inherent step in such IPC computational implementations.

A number of studies also combined IPC with other metrics or incorporated it with existing methods to gain better denoise performance. Huang et al. used the weighted sum of IPC of an input image and its intensity normalized version to detect noise in iris images [94]. Zhu et al. proposed to reduce speckle noise in ultrasound images by incorporating IPC and a feature asymmetry metric into the regularization term of optimization to distinguish features and speckle noise [95]. Luo et al. presented a nonconvex low rank model with IPC and overlapping group sparsity regularization for removing mixed Salt and Pepper noise and random value impulse noise in natural images to preserve local irregular structures [96]. Huang et al. used IPC to replace image gradients in the classic Perona–Malik anisotropic diffusion model to obtain improved edge-preserving noise removal in natural images [97]. Similarly, Gharshallah et al. also constructed a new filter based on a modified anisotropic diffusion combined with the IPC, which is incorporated in the diffusion function to enhance image edges while eliminating noise and texture background for lung CT images [98]. 

IPC has also been combined with an encoder–decoder neural network to reduce noise in low-resolution ultrasound images [18]. The encoder–decoder structure is also inherently multiscale and matches the characteristics of IPC.

### 4.2. Image Quality Evaluation Using IPC

The primary objective of Image Quality Evaluation (IQA) is to assess the quality of images by analyzing their characteristics and evaluating their overall quality [99]. There are two primary aspects of image quality, i.e., fidelity and intelligibility. Fidelity refers to the degree of deviation between the evaluated image and a standard or reference image. Intelligibility focuses on the ability of an image to provide information to humans and/or machines. IQA can be categorized as full-reference, reduced-reference, and no-reference methods based on the availability of reference images. There have been a large number of IPC-based image quality metrics proposed in the literature.

#### 4.2.1. Full-Reference IQA

Liu et al. used the cross-correlation between the IPCs of two images to measure the similarity by dividing them into sub-blocks initially. It showed good sensitivity to various distortions such as noise, mean shift, contrast stretching, blur, and compression [100]. Zhang et al. proposed a Feature Similarity Image Quality (FSIM) index for full-reference IQA where IPC acts as the primary feature and the image gradient magnitude as the secondary feature. After obtaining the local quality map, IPC was again used as a weight function to derive a single quality score. Experiments on six benchmark IQA databases showed that FSIM achieved much higher consistency with subjective evaluations than other state-of-the-arts metrics [101,102,103,104,105].

Instead of using IPC as is, other studies made changes to how phase congruency information is employed or computed. Saha et al. utilized phase deviation-sensitive energy features instead of final IPC to form energy maps, which essentially set the noise compensation term in Equation (19) to a constant as noise should not be removed from IQA [106]. Recently, Zhang et al. proposed a symmetric phase congruency metric, which utilized the sign responses of neighboring pixels to find the symmetry phase congruency to evaluate image quality. They experimentally showed this modification is more sensitive to image structures and more robust to noises [107]. Chen et al. went further by replacing the log-Gabor filters in IPC by Gaussian functions and Laplacian of Gaussian signals to reduce the computational cost while improving performance [108].

#### 4.2.2. Reduced-Reference and No-Reference IQA

Liu et al. proposed a reduced-reference IQA model by applying fractal analysis to IPC to extract features to construct a quality metric [9]. Many more studies were carried out on no-reference IQA using IPC.

Hassen et al. suggested a simple sharpness measure where sharpness is identified as strong local phase congruency, which correlates well with subjective quality evaluations [109]. Other researchers have adopted more sophisticated approaches in using IPC for no-reference IQA by applying some additional operations to IPC and/or combining IPC or IPC-based features with other inputs derived from the input image. Li et al. employed the mean and entropy of the IPC of an image and its mean gradient and entropy as inputs to a regression neural network to perform no-reference IQA that does not require training with subjective evaluations [110,111]. Zhao et al. fed the curvelet features of IPC and its local spectral entropy to a Support Vector Machine (SVM) to train an IQA prediction model [112]. Similarly, Miao et al. combined image gradients and the local binary pattern feature of its IPC as inputs for SVM training [113].

IPC based no-reference IQA for hyper-spectral images was also successfully demonstrated by Shao et al. to have excellent correlation with subjective image quality scores; they used IPC to obtain noise and blur characteristics for each single band in the hyper-spectral input [114].

A special-use case of no-reference IQA is for image reconstruction in computational imaging. The cost function for image reconstruction is essentially an image quality metric. The cost functions in computational imaging are usually defined on simple distance metrics [115]. Tian et al. proposed to construct cost functions using IPC for such applications and showed it can improve the quality of reconstructed images in some lensless imaging systems [116]. Figure 7 shows some image examples using the regularized mean square difference and the IPC-based cost function as the optimization objectives, respectively.

### 4.3. Autofocus and Blur Detection Using IPC

Autofocus is an import function in cameras and other imaging systems to capture high-quality images. Many digital imaging systems utilize image-based methods to achieve automatic focusing, termed passive autofocus [118]. It involves analyzing key features or contrast of an image to automatically adjust the lens, ensuring that the target is clearly imaged. One of the key issues in passive autofocus is blur detection, that is, how to judge whether an image is focused, which is usually accomplished by focus measures [119]. It should be noted that image blur degrades image quality, and thus some of the IQA methods inherently contain blur detection. In this section, blur detection is considered without taking into account other image quality factors.

Blur can disrupt the phase coherence in images; thus, IPC can be used for blur detection [10]. Tian et al. constructed an IPC-based focus measure and compared it with a number of commonly used focus measures derived from image variance, image gradient, image spectrum, and wavelet band ratio. The IPC-based focus measure is more robust for noisy imaging sensors in varying illuminations and has a great balance of defocus sensitivity and effective range [11], as illustrated in Figure 8. Tawari et al. presented a no-reference blur detector based on the statistical features of IPC and image gradient. Blur detection is achieved by approximating the functional relationship between these features using a feed-forward neural network [120,121]. Similarly, Liu et al. combined IPC and image gradient, weighed by a saliency map, to estimate image blur [122].

It should be noted that autofocus systems typically need to run in real time. The computational complexity of IPC-based blur detection is a significant burden for low-cost image systems, especially for consumer electronics.

### 4.4. Image Super-Resolution Using IPC

Image super-resolution aims to produce high-resolution images from low-resolution inputs. It utilizes algorithms and models to recover missing high-frequency details from low-resolution images, resulting in clearer and more detailed ones.

Wong et al. formulated image super-resolution as a constrained optimization problem using a third-order Markov prior model and adapted the priors using the phase variations of low-resolution mammograms [123]. Diskin et al. computed the phase congruency of each pixel’s neighborhood and produced nonlinearly interpolated high-resolution images for dense 3D reconstruction [124]. Zhou et al. proposed a complexity reduction method in multi-dictionary-based super-resolution using IPC. The IPC of a low image is extracted and binarized to distinct the importance of individual image patches. Important high-resolution patches are reconstructed by multi-dictionary-based super-resolution and the unimportant ones by single-dictionary method [125,126]. Nayak et al. put forward a regularization-based super-resolution method by imposing two regularization constraints of structural regularization and high frequency energy. Both terms are computed from IPC. Gradient descent method optimizes the regularized cost function [127].

IPC has also been combined with an encoder–decoder neural network to obtain high resolution ultrasound images from low-resolution ones [18].

### 4.5. Image Watermarking and Slicing Detection Using IPC

IPC has been used to detect the local feature regions of an image, and then a watermark is infused into it using different ways, such as adaptive alpha–beta blending [128,129,130]. Nayak et al. developed an adaptive digital watermarking algorithm in multi-parametric solution space for hiding the copyright information by means of IPC and singular value decomposition-supported information hiding technique [131].

Chen et al. proposed a scheme extracting image features from moments of wavelet characteristic functions and IPC for image splicing detection [132]. Uliyan et al. constructed an image forgery detection algorithm relying on blur metric evaluation and IPC [133]. Hansda et al. presented a hybrid copy–move image forgery detection method using phase adaptive spatio-structured SIFT [134] and the histogram of oriented IPC to localize forgery regions in the presence of intermediate and post-processing attacks [135]. In a more sophisticated vision-based document security system where the content and location of alterations can be detected, IPC was adopted at various stages of the pipeline [136].

## 5. Applications of Phase Congruency in Mid-Level Computer Vision

### 5.1. Feature Detection Using IPC

The definition of a feature can vary depending on the specific problem or application. In general, a feature is a distinctive or interesting part of an image. A crucial aspect of feature detection is its reproducibility; that is, the extracted features should be consistent across different images of the same scene. However, this is not without challenges due to noise, geometric and photometric variations arising from image sensors, and perspective and illumination variations. The origination of IPC was closely related to feature detection, especially due to its space-scale and contrast-invariant nature. It has a significant advantage over gradient-based methods as it is a dimensionless quantity invariant to changes in image brightness or contrast [2,3,7].

#### 5.1.1. Edge Detection

A simple threshold value can be applied to the IPC map to obtain edges over a wide class of images, as shown in Kovesi’s seminal work [3,7,8]. Figure 9 shows an example of using such simple threshold-based IPC edge detection in comparison with several widely used traditional edge detectors, i.e., Sobel, Laplacian of Gaussian, Zero-crossing, and Canny [4,5,6,137].

More sophisticated IPC-based edge detectors have been proposed. Xiang et al. combined ratio-based edge detector and IPC-based edge features as a SAR phase congruency (SAR-PC) edge detector [16]. Shi et al. proposed a conformal monogenic phase congruency-based edge detector that has a good analytical capability in the spatial domain for local structural features [138]. Yang et al. studied edge detection using modified differential IPC [139]. Huang et al. presented an edge detector using the point flow method based on the fusion of multi-scale phase congruency [140].

It should be noted that line detection has both similarities and differences with edge detection. In digital images, a line has two edges. Obviously, we can detect lines by two-stage methods, that is, detecting edges first and then employing post-processing to obtain lines. However, with the proper scales, IPC can directly extract lines in images in one step [141]. This can be seen in Figure 10, particularly obvious for the window structure on the left and the door on the right, where IPC edge detection produces lines while the other detectors mostly generate more complex edges.

#### 5.1.2. Corner Detection

Similar to edge detection, Kovesi also used simple thresholding of the minimum moment of phase congruency for corner detection. Phase congruency was calculated independently in multiple orientations; its moments and their variations with orientations were obtained [8].

Figure 10 shows examples of such simple IPC corner detectors outperforming Harris and SURF detectors [142,143]. It should be noted that for this type of image with significant noise, geometric distortion, and/or non-uniform illumination, a few other widely utilized feature detectors, such as SIFT, ORB [144], and BRISK [145], behave similar to SURF. The robustness of such an IPC-based corner detector was utilized for visual servoing in robotic grasping applications [146].

#### 5.1.3. Ridge Detection

Similar to line, ridge is also a compound feature that is not simply an edge. Micheal et al. proposed a method for automatic ridge detection in lunar images using phase symmetry and phase congruency [147]. Schnek et al. detected ridges, among other sophisticated features, by adjusting the scales of IPC to the proper value [141]. Rensenhofer et al. further developed a ridge detector that exploits the symmetry properties of directionally sensitive analyzing functions in multiscale systems in the framework of alpha-molecules, which is based on IPC [148].

### 5.2. Image Segmentation Using IPC

#### 5.2.1. Image Binarization

Image binarization is a special case of segmenting an image as foreground and background, which is widely utilized in certain image processing applications, such as Optical Character Recognition (OCR).

Tian et al. calculated IPC for a camera-captured document image, from which connected component analysis was carried out to segment out a local window for each symbol; afterwards, a local threshold was calculated for each window to binarize the corresponding grayscale image patch (results illustrated in Figure 11a) [149]. Nafchi et al. initially also employed IPC to select Regions of Interest (ROI) of a document’s foreground [150], but later utilized IPC in more sophisticated fashion by combining the maximum moment of phase congruency covariance, a locally weighted mean phase angle, and a phase-preserved denoised image for ancient document binarization [151,152]. More recently, Bhat et al. proposed a model consisting of phase congruency and a Gaussian model for background elimination using the expectation maximization algorithm in inscription image preprocessing [153].

#### 5.2.2. Image Segmentation

Tian et al. utilized IPC maps to help segment text and non-text areas of the grayscale document images [154]. Li et al. applied a contour-based method to IPC for object segmentation, in particular to reduce the impact of uneven illumination [155]. Figure 11b shows an example of such segmentation.

For medical image processing, Amin et al. employed IPC to segment blood vessels from fundus images [156]. Mapayi et al. combined IPC with fuzzy C-means and gray level co-occurrence matrix sum entropy for the segmentation of retinal vessels [12]. Azzopardi et al. fed IPC of ultrasound images into a Deep Neural Network (DNN) for carotid artery segmentation [157]. Sethi et al. used an edge-based and phase-congruent region enhancement method to segment cancerous regions in liver [158]. Szilagyi et al. utilized an IPC-based feature map to drive level-set segmentation for brain tumors in MRI images [159].

For remote sensing, Wang et al. proposed a high-resolution image segmentation method combining phase congruency with local homogeneity [160]. Zhang et al. used IPC for glacial lake segmentation in the Himalayas in SAR images [161].

### 5.3. Image Matching and Registration Using IPC

#### 5.3.1. Same-Mode Image Registration

Zhang et al. proposed a registration method for images with affine transform relationships. They first used the Maximally Stable Extreme Region (MSER) pairs to carry out a course registration, then IPC-based feature points were obtained from the coarsely aligned image pairs to conduct another step of fine registration [162]. Fan et al. utilized the combination of SIFT, nonlinear diffusion, and IPC for SAR image registration, where IPC was mainly used for removal of erroneous key points [162]. Ma et al. used IPC and spatial constraints for SAR image registration [163]. Dalvi et al. utilized IPC feature points and Iterative Closest Point (ICP) matching for slice-to-volume MRI image registration [164].

#### 5.3.2. Multi-Modal Image Registration

Yu et al. computed an oriented magnitude histogram and the orientation of the minimum moment of an IPC-based histogram and normalized and concatenated the two histograms as feature descriptors for general purpose multi-modal image registration [165].

For remote sensing, Ye et al. used IPC for registering optical and SAR images as well as LiDAR data [166]. Fan et al. combined nonlinear diffusion and phase congruency structural descriptors for optical and SAR image registration [167]. For the same purpose, Wang et al. proposed a uniform Harris feature detection method based on multi-moment of the IPC map and a local feature descriptor based on the histogram of IPC orientation on multi-scale amplitude index maps [168]. Xiang et al. proposed an improved IPC model specifically for SAR images while using the traditional IPC for optical image feature extractions [169]. Hu et al. presented a multispectral line segment matching algorithm based on IPC and multiple local homographies for image pairs captured by cross-spectrum sensors [170]. Further improvement has been explored by more sophisticated feature extractions from IPC or the intermediate output of IPC calculations for optical and SAR image registrations [23,171,172].

For biomedical imaging, Xia et al. applied a SIFT detector on IPC and coherent point drift for multi-modal image registration [173]. For the same purpose, Zhang et al. combined regional mutual information and IPC to utilize both structural and neighborhood information to obtain more robust and higher accuracy [174]. Later, a more sophisticated approach was developed. A Local Phase mean and Phase Congruency values of different Orientations (LPPCO) using filter banks at different orientations and frequencies are first computed. Then, a similarity measure using the normalized cross correlation (NCC) of the LPPCO descriptors is obtained, followed by a fast template matching for detecting correspondences between different images [174].

### 5.4. Image Fusion Using IPC

#### 5.4.1. Same-Mode Image Fusion

Zhan et al. used IPC for multi-focus image fusion, where IPC was essentially used to replace focus measures to be more robust to noise [175]. Mei et al. decomposed images into a base layer and a detail layer and used total variation and IPC for the two layers, respectively, for multi-focus image fusion [176]. For the same purpose, Yazdi et al. proposed to combine IPC and Non-Subsampled Contourlet Transform (NSCT) [177,178]. Asadi et al. carried out multi-exposure image fusion via a pyramidal integration of IPC with the intensity-based maps [179].

#### 5.4.2. Multi-Modal Image Fusion

Zhao et al. made use of IPC and its moments comparing the local cross-correlation of corresponding feature maps of input images and fused output to access its quality without a reference [180]. Zhang et al. proposed a multi-modal image fusion algorithm with the shiftable complex directional pyramid transform, where phase and magnitudes of complex coefficients are jointly considered [181].

For remote sensing, Huang et al. combined IPC and NSCT with entropy for fusing infrared and visible images [14]. For the same purpose, Liu et al. used IPC and a guided filter in Non-Subsampled Shearlet Transform (NSST) to fuse the images. And Wang et al. introduced quadtree decomposition and Bezier interpolation to extract crucial infrared features and proposed a saliency advertising IPC-based rule and local Laplacian energy-based rule for low- and high-pass sub-band fusion [182]. Chen et al. employed saliency detection and Gaussian filters to decompose source images into salient, detail, and base layers. Furthermore, they adopted a nonlinear function to calculate the weight coefficients to fuse salient layers and used an IPC-based fusion rule to fuse the detail layers so that the details could be retained better [183]. Fu et al. utilized IPC and a simplified Pulse-Coupled Neural Network (PCNN) as a basic fusion framework using the generalized intensity–hue–saturation transform and NSCT for SAR and optical image fusion [184]. Ye et al. presented an illumination-robust subpixel Fourier-based image correlation method using IPC. Both the magnitude and orientation information of IPC features were adopted to construct a structural image representation, which is embedded into the correlation scheme of the subpixel methods, either by linear phase estimation in the frequency domain or by kernel fitting in the spatial domain, achieving improved subpixel methods [185]. More recently, Fan et al. constructed pyramid features of orientated self-similarity for multi-modal remote sensing image matching, which integrates IPC into the self-similarity model for better encoding structural information [22]. They also proposed a modified uniform nonlinear diffusion-based Harris detector to extract local features, which employs IPC instead of image intensity for feature extraction and thus obtains well-distributed and highly repeatable feature points. A local structural descriptor, namely IPC order-based local structure, was designed for the extracted points [172].

For biomedical imaging, Dhengre et al. decomposed CT and MRI images into low and high frequency sub-bands by NSCT. The low frequency sub-bands were processed to extract IPC image features; the details were extracted from the high frequency sub-bands by using a guided filter to preserve the edge details [186]. Bhatnagar et al. also transformed the source images by NSCT and used two different fusion rules based on IPC and directive contrast to fuse low- and high-frequency coefficients [187]. Similarly, Zhu et al. presented a multi-modality medical image fusion method using IPC and local Laplacian energy. NSCT was first performed on image pairs to decompose them into high- and low-pass sub-bands. The high-pass sub-bands were integrated by an IPC-based fusion rule to enhance the detailed features; a local Laplacian energy-based fusion rule was used for low-pass sub-bands [188]. Arathi et al. used slantlet transform and IPC for CT and MIR image fusion [189]. Tang et al. proposed an IPC-based green fluorescent protein and phase contrast image fusion method in NSST. The source images were decomposed by NSST to multiscale and multidirection representations. The high-frequency coefficients are fused with IPC and parameter-adaptive PCNN, while the low-frequency coefficients are integrated through a local energy-based rule [190].

## 6. Applications of Phase Congruency in High-Level Computer Vision

### 6.1. Object Detection, Tracking, and Recognition Using IPC

Object detection, tracking, and recognition are often intertwined, and as such, we discuss them together.

Tian et al. proposed a simplified IPC computation scheme and applied it to detect endotracheal tubes in X-ray images [191]. Sattar et al. also used IPC for tooth detection in dental radiographs [192]. Rahmatullah et al. employed local features from an intensity image and global feature symmetry from its IPC for the detection of the stomach and the umbilical vein in fetal ultrasound abdominal images [193]. Verikas et al. combined IPC-based circular object detection, stochastic optimization-based object contour determination, and SVM as well as random forest classifications for *Prorocentrum minimum* cell recognition in phytoplankton images [15]. Teutsch et al. used IPC for automated recognition of bacteria colonies, as well as coded markers for both 3D object tracking and automated camera calibration [194].

Santhaseelan et al. developed a robust method to track objects of low resolution in wide-area aerial surveillance imagery using IPC and derived rotation invariant features [195]. Zhang et al. introduced IPC-based on sub-aperture coherent method and the differences of texture feature in sub-aperture to realize target detection in SAR images [196]. Zhang et al. computed three local features under the IPC framework using a set of quadrature pair filters and integrated them by score-level fusion to improve finger–knuckle–print recognition accuracy [197].

Not surprisingly, a number of face recognition algorithms have utilized IPC for its characteristics of being insensitive to illumination and image contrast variations. Bezalel et al. combined IPC and Gabor wavelet for efficient face recognition [198]. Basavaraj et al. achieved improved face recognition using neighborhood-defined modular IPC-based kernel principle component analysis [199]. Hamd et al. combined IPC, gradient edges, and their associate angles for face classification [200]. Essa et al. used local directional patterns of IPC for illumination-invariant face recognition [201,202]. While Koley et al. achieved illumination-invariant face recognition using a more sophisticated fused cross-lattice pattern of IPC [203].

### 6.2. Other High-Level Applications Using IPC

Cinar et al. utilized IPC to detect and quantify cracks and openings in quasi-brittle (granitic rock) and ductile (aluminium alloy) materials [93,204]. Deng et al. came up with iterative IPC for 3D volume CT images for crack detection, which is particularly effective to detect radially distributed cracks in cylindrical objects [205].

Bucie et al. introduced an IPC-based method to extract the facial features for facial expression recognition. It computes the IPC between the images, and analysis is performed in the frequency space where the similarity between image phases is measured to form discriminant features [206]. Others have extracted different features, such as binary patterns, from IPC for facial expression recognition [207,208].

## 7. Challenges for Practical Applications of IPC

As discussed above, there are many advantages of IPC, especially its insensitivity to image scale and orientation, image contrast, and illumination conditions. However, for wider adoption for practical applications, there are still a number of challenges.

### 7.1. Noise Sensitivity

In noisy images, IPC can lead to inaccurate or spurious feature detection. Though the impact of noise can be mitigated to a certain degree by subtracting a noise threshold in IPC computation as is typically done, doing so inevitably reduces the IPC sensitivity to high-frequency details in images. When noise reaches a significant level, as happens in active thermography images [209], this approach may fail to distinguish noise from high-frequency image features, especially when the number of scales is small. As demonstrated below, in the presence of heavy noise in thermograms, IPC performed much better in Figure 12 compared to Figure 13, where the noise is more severe, though the images were from the same application scenario.

### 7.2. Computational Complexity

Due to the multi-scale and multi-orientation nature of IPC computation, its computational cost via log-Gabor filter banks is very high, especially when large numbers of scales and orientations are used. Though the adoption of monogenic filters for computing IPC can alleviate the computational complexity to a certain degree [86,87], it is still not as simple as some other image transformations. This is a significant issue when dealing with large-size images or real-time applications. Furthermore, reduced IPC computations from monogenic filters come at the cost of reduced scale feature maps, which have been utilized in various IPC-based applications. Reducing the number of scales and orientations can improve the computational efficiency significantly, but at the cost of low IPC accuracy, as happens in any approximation problems (see examples in Figure 12 and Figure 13). Computational time is compared for different scales and orientations using image Figure 11a and its up-sampled version using Kovesi’s IPC implementation in Matlab 2023a. Computations were carried out in single thread on a personal computer using a single core of a 12th Gen Intel i7-12700K 3.6 GHz CPU with 16G memory, and the results are shown in Figure 14. Generally speaking, computational time increases linearly with the number of either scales or orientations, as well as the number of pixels in the image. As the number of pixels is the power function of resolution and dimension in images, IPC computational time can increase dramatically for high-resolution and high-dimension ones.

### 7.3. Parameter Tuning

IPC computation algorithms have multiple parameters that need to be adjusted for optimal performance. Determining the best parameter settings can be challenging and often requires extensive experimentation and fine-tuning. Though improvement has been made, it is still not fully resolved [71,210]. For example, in Figure 12, if we only consider the red rectangular ROI, a scale of 8 and orientation of 2 produce the cleanest IPC (i.e., the impact of noise is alleviated); when taking into other regions in the image, a scale of 6 and orientation of 4 probably produce the overall best IPC. In Figure 13, none of the combinations of scales and orientations produces very clean IPC; the overall best IPC is likely to be at scale 6 and orientation 2. Increasing the orientation number actually leads to some undesirable artifacts in IPC.

### 7.4. Integration with Other Image Features

In many cases, it may be desirable to combine IPC with other image features, such as color, texture, etc., to improve the performance of vision algorithms. However, effectively and efficiently fusing these features is challenging and requires considering their complementarity and redundancy [211]. By and large, such integration is still carried out case by case, i.e., feature engineering, based on the user’s domain expertise, as described in numerous examples in Section 4, Section 5 and Section 6.

## 8. Potential Improvement of IPC Using Deep Learning

As mentioned in previous sections, DNNs have been combined with IPC in various application studies. IPC has been combined with an encoder–decoder neural network to reduce noise in low-resolution ultrasound images and achieve super resolution [18]. IPC and its derived quantities have been employed as inputs to a regression neural network to perform no-reference IQA that does not require training with subjective evaluations [110,111]. Blur detection has been achieved by approximating the functional relationship between the statistical features of IPC using a feed-forward neural network [119,120]. IPC of ultrasound images has been fed into a deep neural network for carotid artery segmentation [157]. It has also been utilized together with a simplified PCNN as a basic fusion framework using the generalized intensity–hue–saturation transform and NSCT for SAR and optical image fusion [184]. A fused cross-lattice pattern of IPC has been used as input to a lightweight CNN for illumination-invariant face recognition [203]. These studies used IPC and/or its derivatives as inputs to some DNNs, and such integrations are not very tight; that is, IPC computation and DNN training are essentially separate and can be easily decoupled.

In recent years, DNNs have also been utilized to implement some traditional image transforms; this is very relevant to IPC as it is essentially a nonlinear image transform. For example, Guan et al. proposed an interpretable wavelet DNN integrating multi-resolution analysis at the core of its design. By using a lifting scheme, it can generate a wavelet representation and a neural network capable of learning wavelet coefficients in an end-to-end form [212]. And a number of other studies have come up with different ways to implement wavelet DNNs [213,214,215].

As the most widely employed IPC implementation is based on log-Gabor wavelets, we believe it is feasible to extend some of the aforementioned wavelet DNNs to compute IPC. And such IPC-DNNs can be concatenated or combined in parallel with other network modules for various computer vision applications. This approach is illustrated in Figure 15 using the widely used encoder–decoder UNet architecture [216,217]. In this paradigm, it is not difficult to build in noise removal schemes in the network adaptable to different noise levels and characteristics. In this framework, IPC computations are built into the DNN structure and may lead to more efficient and effective computer vision application pipelines than those simply concatenating them, as discussed above.

In a simpler approach, we can also utilize machine learning to learn the appropriate parameters for IPC computation for higher efficiency and better performance at runtime. For example, for specific applications, the optimal numbers of scales and orientations, as well as other parameters, can be learned in advance using the cost functions defined with desired application outputs. The structured forests for edge detection method can be viewed as a good example of this approach [218].

## 9. Conclusions

Image Phase Congruency (IPC) is closely connected to how the human visual system interprets and processes spatial frequency information. In this survey, evidence from biological vision research that supports the use of IPC by the human perceptual system is introduced. Next, the fundamental mathematical model and different computational implementations of IPC are described and compared. Various applications of IPC in computer vision, from low-, mid-, to high-level vision, have been comprehensively summarized and categorized. Multiple graphical examples are presented for some of these applications to highlight the benefits of using IPC. The current challenges in implementing IPC in practice and potential ways to improve its effectiveness using deep learning are also briefly discussed. This review is expected to introduce IPC to a wider audience and foster future work to overcome its limitations and expand its applications.

## Figures and Tables

**Figure 1 biomimetics-09-00422-f001:**
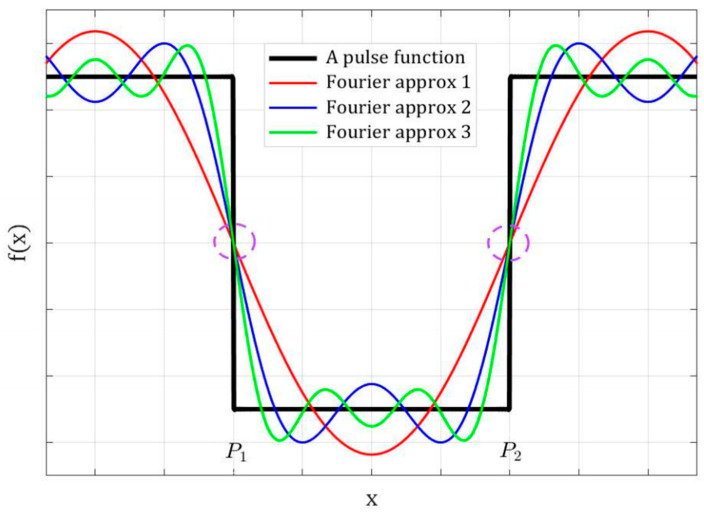
A pulse function and its approximations by different numbers of Fourier components.

**Figure 2 biomimetics-09-00422-f002:**
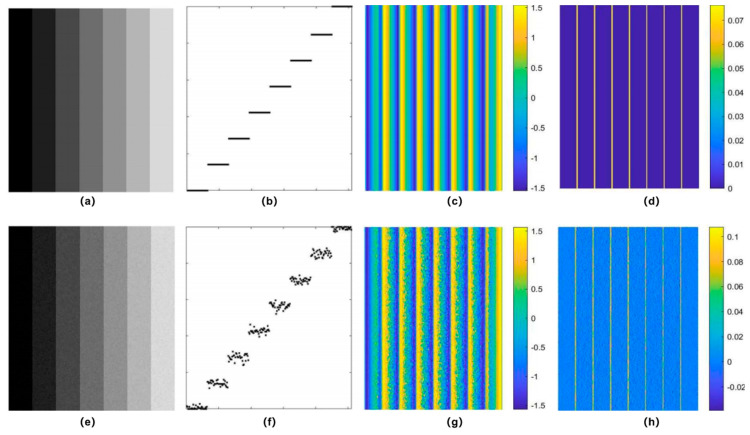
Mach bands. (**a**) An ideal step intensity pattern; (**b**–**d**) Intensity cross-section, weighted local phase, and horizontal gradients of (**a**). (**e**) A step intensity pattern with Gaussian noise. (**f**–**h**) Intensity cross-section, weighted local phase, and horizontal gradients of (**e**).

**Figure 3 biomimetics-09-00422-f003:**
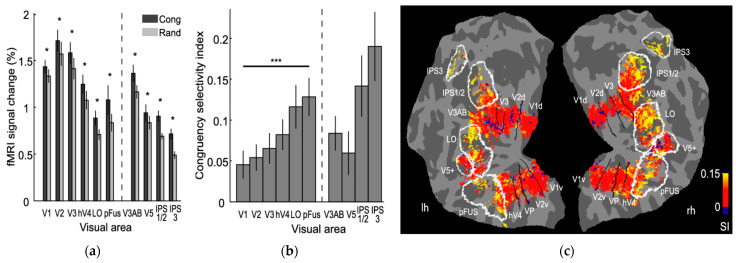
Results of an fMRI study showing human visual cortex is tuned to phase congruency. (**a**) Mean fMRI signal changes for the congruent (**Cong**) and random (**Rand**) stimulus category blocks shown for regions of interest in various visual areas averaged from both hemispheres across all subjects. Asterisks indicate statistically significant differences between conditions (* *p* < 0.05); Wilcoxon signed rank test. (**b**) Congruency selectivity indices for the visual areas (*** *p* < 0.001). (**c**) Group-averaged congruency selectivity index maps for left and right hemispheres (reproduced with permission from Henriksson et al. [50]; Copyright 2009 Society for Neuroscience).

**Figure 4 biomimetics-09-00422-f004:**
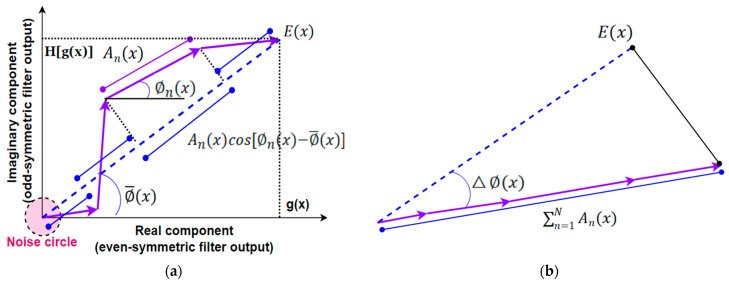
Graphical illustration of the phase congruency of a signal, its local energy, and the amplitude sum of the Fourier components. (**a**) Total local energy (dashed blue line) and individual Fourier components as complex vectors (purple arrows) and their relationships. (**b**) Phase congruency inequality (the sum of the magnitudes of individual Fourier components is greater than or equal to the total local energy, where “equal to” happens if and only if phase angles of all components are perfectly aligned, that is, the average phase deviation △∅(x) is zero).

**Figure 5 biomimetics-09-00422-f005:**
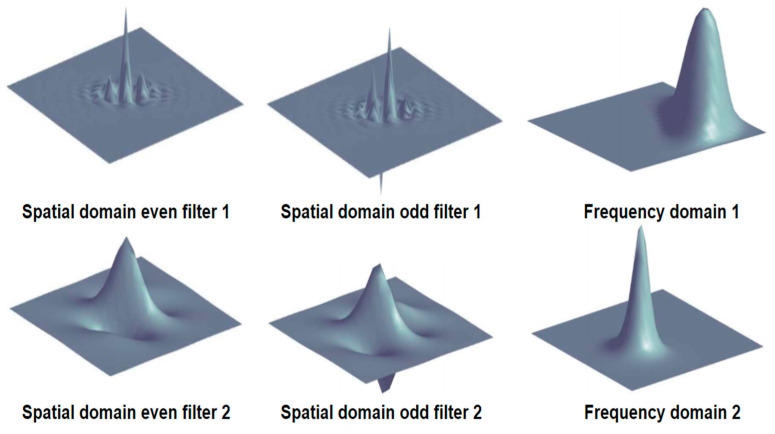
Graphical illustrations of log-Gabor filters in the spatial and frequency domains.

**Figure 6 biomimetics-09-00422-f006:**
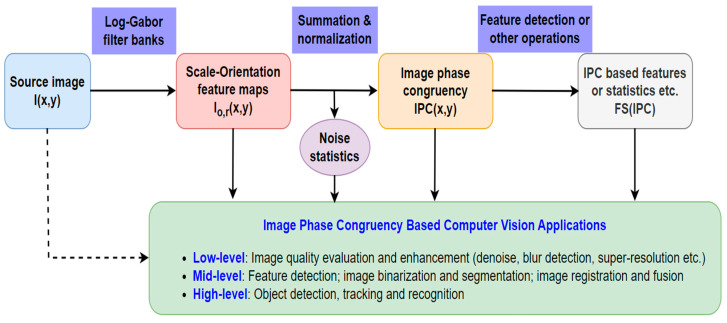
Illustration of various approaches to IPC applications. IPC statistics are statistical values computed from IPC, such as average, standard deviation, percentile, and entropy, etc.

**Figure 7 biomimetics-09-00422-f007:**
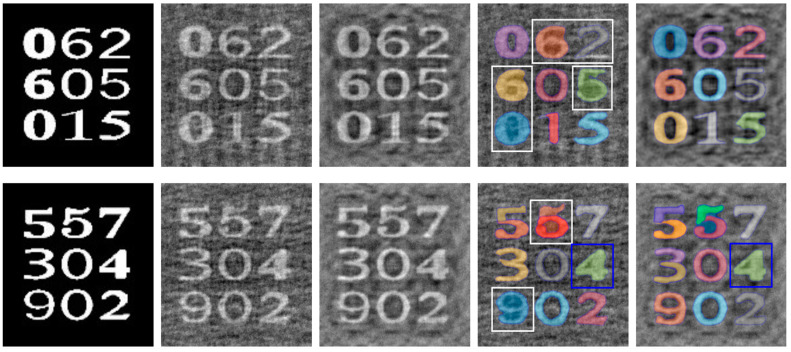
Examples of image reconstruction in lensless imaging by different cost functions. From left to right: ground truth; reconstructed images with regularized mean square difference cost function and IPC-based cost function; Segment Anything Model (SAM) outputs of reconstructed images with regularized mean square difference cost function and with IPC-based cost function. In the last two columns, white boxes indicate failed segmentation in one of two cases, and blue boxes indicate failed segmentation in both cases (the reconstruction method was presented by Tian et al. [116]; SAM was described by Kirillov et al. [117]).

**Figure 8 biomimetics-09-00422-f008:**
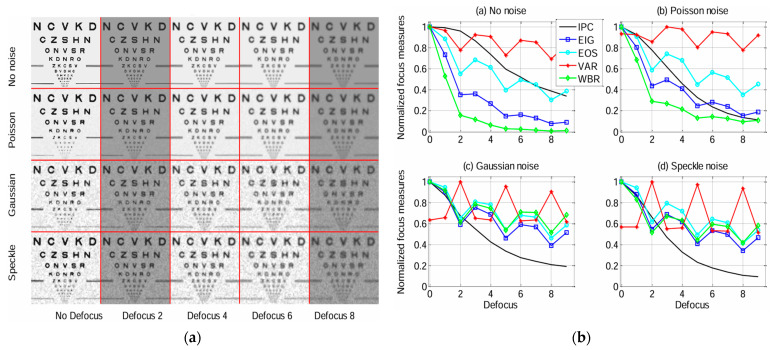
Comparison of focus measures for autofocus. (**a**) Images with different types of noise and varying illumination/contrast. (**b**) Focus measure results for images in (**a**) (EIG: Energy of Image Gradient; EOS: Energy of Spectrum; VAR: Variance; WBR: Wavelet Band Ratio) (Reproduced with permission from Tian et al. [11]; Copyright 2011 Optical Society of America).

**Figure 9 biomimetics-09-00422-f009:**
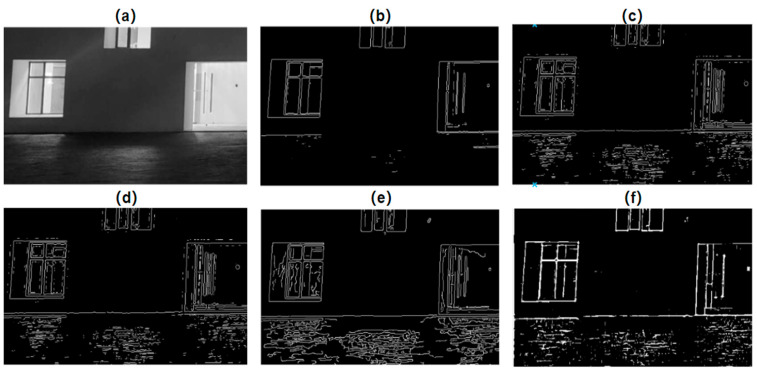
Comparison of various edge detectors. (**a**) Gray image. (**b**–**f**) Edge detection from Sobel, Laplacian of Gaussian, Zero-crossing, Canny, and IPC, respectively (Sobel, Laplacian of Gaussian, Zero-crossing, and Canny all used built-in functions in Matlab 2023 with default parameters, and IPC also used default parameters in Kovesi’s implementation, and a fixed threshold of 0.2 was applied to the IPC to obtain the edge map).

**Figure 10 biomimetics-09-00422-f010:**
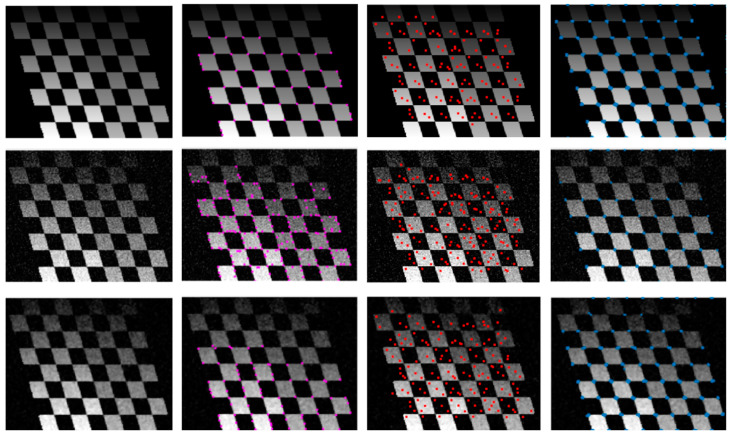
Comparison of various corner detectors. From top to bottom, image with no noise, image with Gaussian noise, and noisy image after median filtering. From left to right, gray image, results from Harris detector (purple dots), SURF detector (red dots), and IPC-based corner detector (blue dots) (Harris and SURF detectors used built-in functions in Matlab 2023 with default parameters, and IPC also used default parameters in Kovesi’s implementation, and a fixed threshold of 0.1 was applied to the minimum moment of IPC to obtain the corners).

**Figure 11 biomimetics-09-00422-f011:**
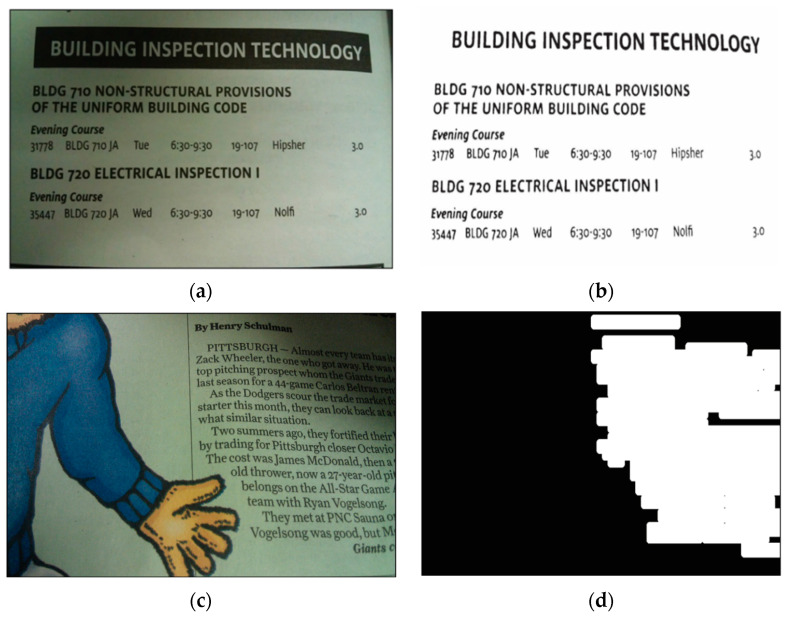
Document image binarization and segmentation examples. (**a**,**c**) Camera-captured document images. (**b**) Binarization results of (**a**). (**d**) Text segmentation results of (**c**).

**Figure 12 biomimetics-09-00422-f012:**
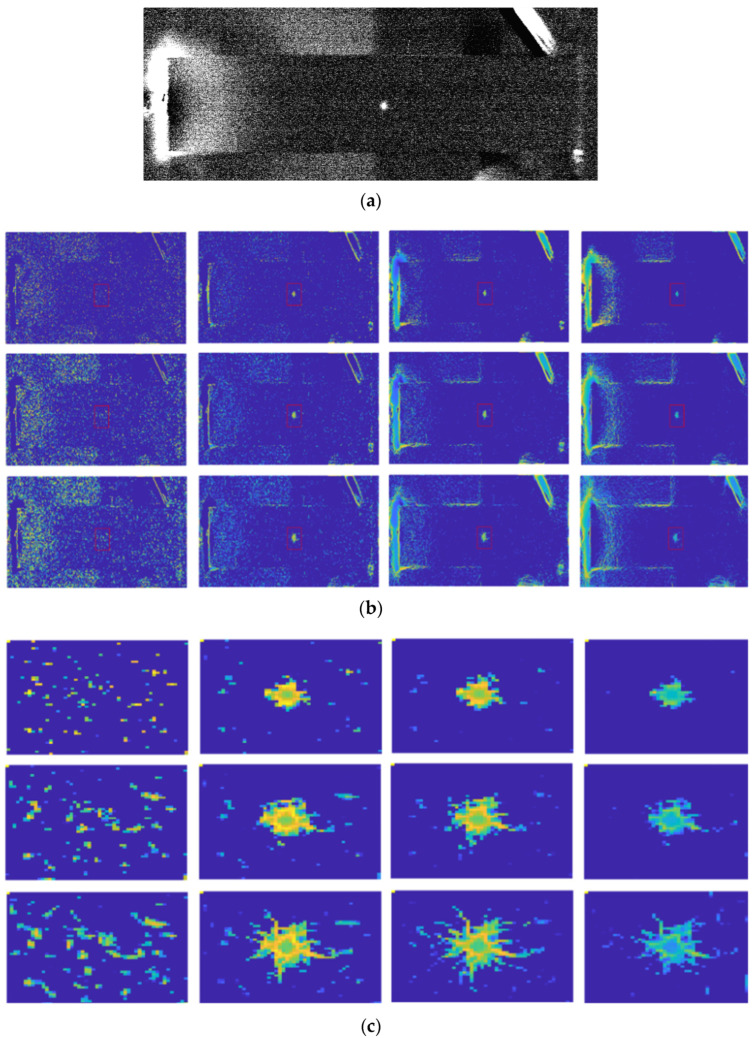
IPC computed with different scales and orientations. (**a**) An active thermography image with significant noise. (**b**) Pseudo-color representations of IPCs computed from (**a**), from left to right: scales of 2, 4, 6, and 8, respectively; from top to bottom: orientation of 2, 4, and 6, respectively. (**c**) Magnified regions as indicated by red rectangles in (**a**), not in proportion.

**Figure 13 biomimetics-09-00422-f013:**
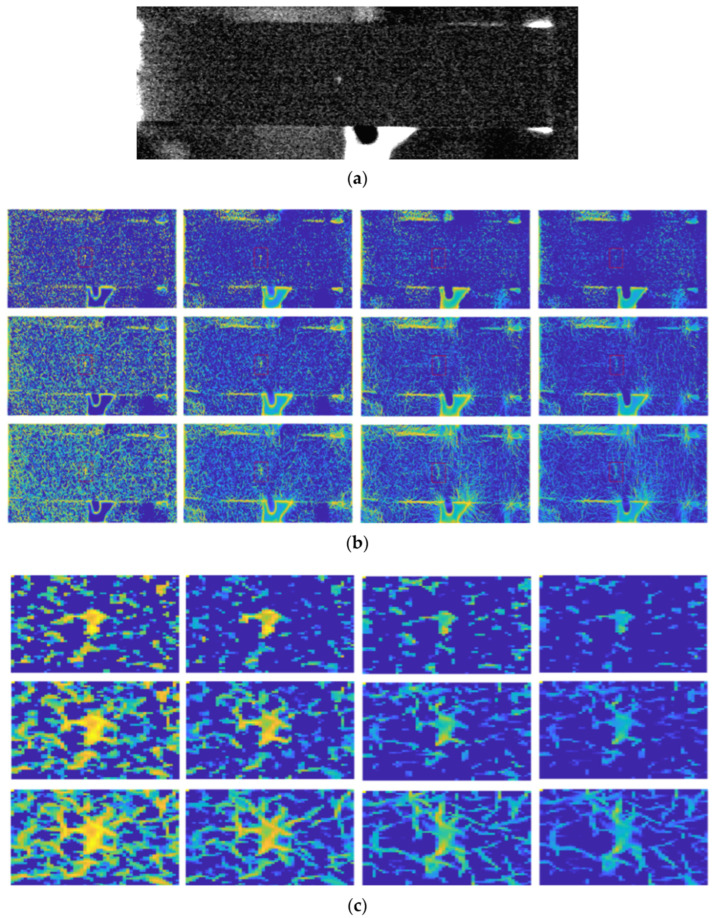
IPC computed with different scales and orientations. (**a**) An active thermography image with heavy noise. (**b**) Pseudo-color representations of IPCs computed from (**a**), from left to right: scales of 2, 4, 6, and 8, respectively; from top to bottom: orientation of 2, 4, and 6, respectively. (**c**) Magnified regions as indicated by red rectangles in (**a**), not in proportion.

**Figure 14 biomimetics-09-00422-f014:**
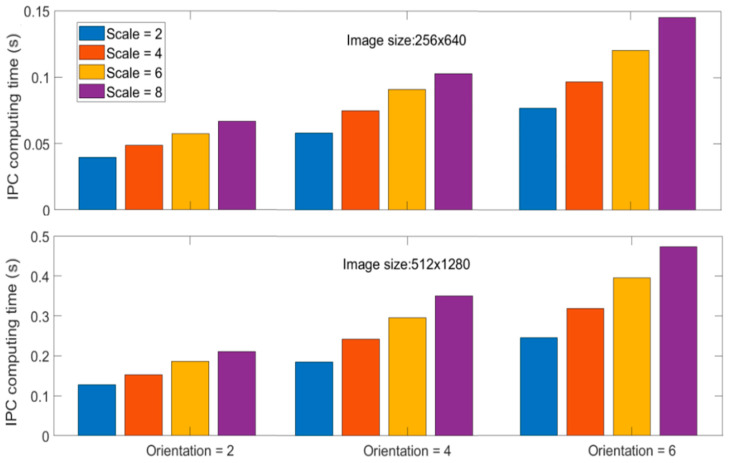
IPC computing time for different scales, orientations, and image sizes. The image of size 256 × 640 is from Figure 11a, and the other image is its up-sampled version. Computations were carried out in single thread in Matlab 2023a on a personal computer using a single core of a 12th Gen Intel i7-12700K 3.6 GHz CPU with 16G memory. Each number is the median of 10 repeated identical runs to reduce the randomness of runtime due to various factors.

**Figure 15 biomimetics-09-00422-f015:**
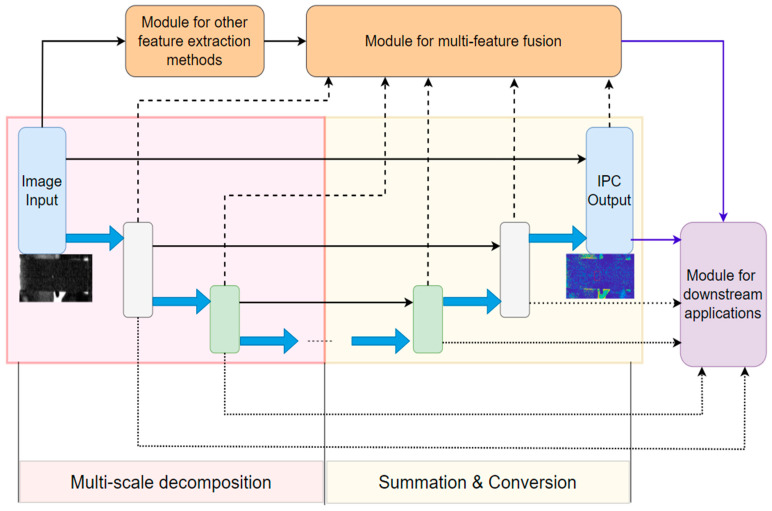
Illustration of the concept of DNN-based IPC implementation and its tight integration with other network modules for various computer vision applications.

## Data Availability

Data will be available upon reasonable request.

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
