# Peer review of "Biological Basis and Computer Vision Applications of Image Phase Congruency: A Comprehensive Survey"

_biomimetics, 2024, doi:10.3390/biomimetics9070422_

Round 1

Reviewer 1 Report

Comments and Suggestions for Authors

The paper presents a survey of Image Phase Congruency methods in Image Processing and Computer Vision. The authors have structured the paper on the level of processing and also made a comprehensive introduction in the field, which I appreciate as didactic.

The paper is well structured, comprises a large number of up-to-date references and illustrates the most performing methods.

The authors have also a personal contribution, testing some of the methods of IPC and as the Deep Learning techniques are a state of the art, they propose an enhancement of the results using DL. I recommend to extend this subchapter, as I will be of interest for the readers. There are also some papers that worth citing. Some DL methods are cited in bulk, [18-24] in line 58, and should be developed in large, in the mentioned chapter.

Figure 14 presents the computational time of IPC, however, it will requires for the authors to describe also the enviromnent used for testing in text, not in the caption of the figure, as a sub-chapter was dedicated to it.

Comments on the Quality of English Language

Some expressions needs to be reviewed.

Page 8, line 293, spell Kovesi instead of Kosevi. Also, in line 804, it appears as "Kevosi". What is the real name of the referenced method author?

Reviewer 2 Report

Comments and Suggestions for Authors

Positive:
- The article is clear and easy to follow, with a well-conducted literature review.
- The article provides a comprehensive and updated survey of the applications of Image Phase Congruency in computer vision, including its potential uses in the perception field.

Negative:
-
The manuscript is overly verbose and seems to rely excessively on generative text tools, impacting the conciseness of the content. Some words are very repeated: pivotal, delve, rooted, employ, constituent,...
- The manuscript needs a detailed discussion on how noise and nonlinear radiometric differences affect IPC. This is important for understanding IPC's robustness in real-world scenarios.
- Figure 6: what is "statistics etc"?? 

Suggestions:
- Missing some references related to applications: Burlacu, Adrian, and Corneliu Lazar. "Image features detection using phase congruency and its application in visual servoing." 2008 4th International Conference on Intelligent Computer Communication and Processing. IEEE, 2008.
- Line 91: Missing space between Fig and 1 ("Fig.1 illustrates a..."). Also in Fig.3.
- Fig. 2 should be shown after the mention. Also it happens with other figures. It is advisable to show the images after their mention.
- Line 111: Lack of space in "to find the maximum of fis equivalent to finding where"
- Missing some reference related to the SAR applications: https://www.mdpi.com/2072-4292/12/20/3339
- Line 455: Typo in "no-reference"?
- Line 155: Typo in "occur"
- Use the same "-" symbol for "space–time " than in "spatio-temporal".
- From line 548 to 553 is in different colour the text.
- Line 804: Typo in the name "Kevosi's..."
- In the high level applications I miss some discussion of potential uses in face alignment and skin segmentation. This article discusses these issues for applications related to the human health sensing: Casado, Constantino Alvarez, and Miguel Bordallo López. "Face2PPG: An unsupervised pipeline for blood volume pulse extraction from faces." IEEE Journal of Biomedical and Health Informatics (2023).

Comments on the Quality of English Language

The article is clear and easy to follow. However, it could be improved as the manuscript is overly verbose and seems to rely excessively on generative text tools, impacting the conciseness of the content. Some words are repeated frequently: pivotal, delve, rooted, employ, constituent. With just a little refinement, it will become a nice and didactic survey.
